# Statistical Insight into China’s Indigenous Diagnosis-Related-Group System Evolution

**DOI:** 10.3390/healthcare11222965

**Published:** 2023-11-15

**Authors:** Wenlong Ma, Jing Qu, Hui Han, Zixia Jiang, Tiantian Chen, Xuefeng Lu, Jiaoyang Lu

**Affiliations:** 1Department of Gastroenterology, Qilu Hospital, Cheeloo College of Medicine, Shandong University, Jinan 250012, China; sdumwl@mail.sdu.edu.cn (W.M.);; 2Department of Medical Records, Qilu Hospital, Cheeloo College of Medicine, Shandong University, Jinan 250012, China; 3Office of Hospital President, Qilu Hospital, Cheeloo College of Medicine, Shandong University, Jinan 250012, China; 4Department of Medical Insurance, Qilu Hospital, Cheeloo College of Medicine, Shandong University, Jinan 250012, China; 5Medical Integration and Practice Center, Cheeloo College of Medicine, Shandong University, Jinan 250012, China

**Keywords:** China, diagnosis intervention packet, diagnosis-related groups, evolution, cholecystitis, payment

## Abstract

The use of Diagnosis-Related Groups (DRG) is a prevalent payment system employed to control hospitalization costs and improve medical efficiency. China has developed an indigenized DRG payment system including Single Disease Payment (SDP), DRGs, and Big Data Diagnosis-Intervention Packet (DIP). In this study, we took cholecystitis as an example, drawing on both primary and secondary data to verify the effectiveness of China’s indigenized DRG system and to introduce the evolution of DRGs in China. Primary data were gathered from Qilu Hospital in 2019–2021. Secondary data were collected from published literature from 2004–2016. Only studies with both pre-SDP/DRG and post-SDP/DRG groups were included. Among the studies included, 92.9% (13/14) reported a significant reduction in hospitalization costs after the implementation of SDP while other studies identified length of stay (LOS) and age as the most significant influential factors in SDP. Furthermore, we elaborated the efficiency of DRGs using data from 2738 inpatients in Qilu hospital. Moreover, 60% (6/10) of the studies from the databases also showed the efficiency of DRGs in different regions. SDP is efficient in saving hospitalization costs, but its implementation is limited. DRGs have a broader scope of application, but their effectiveness remains to be validated. DIP is a brand new concept in China, and more data are needed to assess its efficiency.

## 1. Introduction

Originating at Yale University, Diagnosis-Related Groups (DRGs) were first implemented in the United States in 1983 and are considered a highly effective payment system to improve efficiency, including controlling hospitalization costs and increasing medical efficiency. In this payment system, patients are classified into groups based on their diagnosis and additional characteristics, such as age, complications, and operations, allowing medical expenses to be directly related to the group [1]. Over the past three decades, DRGs have been widely adopted and utilized by several developed countries, including the United States, Australia, Finland, France, Germany, and Japan [2,3,4,5]. Taking Japan, an Asian country, as an example, Hamada et al. evaluated the impact of the diagnostic procedure combination/per-diem payment system (DPC/PDPS) (known as the Japanese DRGs) and discovered that this system could decrease medical expenses. This was evidenced by a decrease in both the average per capita hospitalization cost and the length of stay (LOS) in hospitals [2].

As the world’s most populous developing country, China has made significant progress in providing Universal Health Coverage through three independent insurance systems that cover employed workers, city residents, and farmers, accounting for over 95% of the population [6]. However, this progress has resulted in high health expenditures and has put pressure on the limited national insurance funding in recent years. In response, the Chinese government has been searching for an alternative payment system that can provide essential healthcare services to the majority of the population while reducing spending, and DRG has emerged as a potential solution.

Taking China’s national conditions into consideration, the Chinese government has successively developed a system including three kinds of indigenous DRGs: Single-Disease Payment (SDP, which is also regarded as simplified DRG), DRG, and the recent Big Data Diagnosis-Intervention Packet (DIP). The interplay among SDP, DRG, and DIP can be likened to an intricate trio in the realm of healthcare reform: SDP serves as a succinct yet indispensable component, initiating the prelude of China’s healthcare reform; DRG assumes the role of a classic melody, propelling reform efforts forward; and DIP incorporates ethnic elements to create a harmonious resolution, offering a uniquely Chinese approach. Together, these three components intertwine and reinforce each other, producing a resounding crescendo in China’s ongoing medical reform narrative.

Though this system is marvelous, previous studies only investigated the effect of SDP [7] or DRG [8] alone, but failed to uncover the intrinsic connection between them, nor distinguished their respective roles in reducing cost and increasing medical efficiency. In this study, we took cholecystitis, a common disease of the digestive system, as an example to elaborate the evolution and differentiation of the three indigenous DRGs. Additionally, we analyzed recent studies to determine the effectiveness of these DRGs in reducing expenditures and improving medical efficiency. By providing an overview of the development of DRGs in China and evaluating their benefits and drawbacks at each stage, we aimed to offer insights into building a high-efficiency, cost-effective Universal Public Insurance system in China and other developing countries.

## 2. Materials and Methods

This study utilized both primary and secondary data sources to assess the effectiveness of China’s characteristic DRGs in reducing medical expenses and increasing medical efficiency in hospitals.

### 2.1. Primary Data Collection

Primary data were gathered from Qilu Hospital, a large regional medical center located in a province with a population of over 100 million people. DRG was integrated into the hospital information system on 1 October 2019 and was put into trial operation on 30 September 2020. Based on this timeline, patients with cholecystitis-related disorders admitted to Qilu Hospital between 1 October 2019 to 30 September 2021 were divided into a “pre-DRG” group and a “post-DRG” group. Cholecystitis-related disorders were identified using the primary diagnosis with the International Classification of Diseases, 9th Revision, Clinical Modification of Operations and Procedures, ICD-9-CM-3. The study population included 2738 patients with cholecystitis-related disorders, which were stratified into two groups based on their index data (N = 1172 before the DRG from 1 October 2019 to 30 September 2020; and N = 1566 after the DRG from 1 October 2020 to 30 September 2021). All costs were analyzed based on charge reports from the hospital.

### 2.2. Secondary Data from Published Literature

#### 2.2.1. Search Strategy and Selection Criteria

A comprehensive literature search was conducted in electronic databases including Scopus, PubMed, Web of Science, China National Knowledge Infrastructure Database (CNKI), and Wan Fang Database. The search was conducted using the keywords “Diagnosis-Related Groups” [Mesh], “Single Disease Payment,” “Diagnosis-Intervention Packet (DIP),” “Cholecystitis” [Mesh], and “China” [Mesh], and was limited to articles published between 2004 and September 2022. The beginning year of 2004 was selected because it marked the nationwide implementation of the Single-Disease Payment (SDP) in China. In the assessment of DRG efficiency, the search was further limited to articles published between 2019 and September 2022 following a prior systematic review [8].

#### 2.2.2. Data Extraction

Data extraction was performed by one reviewer (Wenlong Ma) and was independently verified by another reviewer (Jiaoyang Lu). Any discrepancies were resolved through discussion. The extracted data for evaluating the efficiency of SDP and DRG included the first author, publication year, study location, number of pre-SDP/DRG groups, number of post-SDP/DRG groups, length of stay (LOS), and total expenditure (TE). To evaluate the factors affecting the hospitalization cost of cholecystitis, data on the first author, publication year, and logistic regression indicators including length of stay, complications, age, gender, and operation were extracted.

## 3. Results

### 3.1. Study Selection

We initially conducted a literature review of studies examining the efficiency of SDP system for various common diseases, including pneumonia, diabetes, thyroid neoplasms, hypertension, inguinal hernia, cholecystitis, ureteral calculus, cesarean section, and fracture (Figure 1). A heat map was generated based on the results of 254 studies. Given the large sample size and diverse characteristics of the diseases studied, cholecystitis was selected as the focus of our further analysis.

In total, 338 studies were identified that investigated either the impact of the SDP or the factors influencing the cost of treating cholecystitis. After eliminating duplicates, we reviewed the titles and abstracts of 295 articles and excluded 224 of them due to their irrelevant topics, regions, or lack of empirical studies. We then thoroughly read the full text of the remaining 71 citations and finally included 19 publications (13 studies on SDP and 6 studies on the factors influencing cost) in our analysis (Figure 2). Furthermore, from the original search results of 1350 studies, we selected 9 studies for further analysis of the efficiency of DRG (Figure 3).

Among the nine studies evaluated for efficiency of DRG, two were conducted in Guangdong and one each in Henan, Yunnan, Beijing, Zhejiang, Liaoning, Hubei, and Sichuan. All studies included pre-DRG and post-DRG groups. All of them compared the total expenditure, and eight studies included data of LOS (Figure 3).

### 3.2. Characteristics of the Included Publications

The 19 SDP studies included were conducted in different areas of China: three in Shandong, two each in Yunnan, Hunan, Anhui, Jiangsu, Jilin, Hubei, and Beijing, and one each in Liaoning and Tianjin. All studies for efficiency of SDP included post-SDP group (implementing SDP) and pre-SDP group (traditional payment system, not implementing SDP). Fourteen studies reporting total expenditure and LOS were used for evaluating the efficiency of SDP; six studies using logistic regression to predict potential influencing factors such as LOS, complication, age, gender, and operation were pooled and analyzed.

### 3.3. Efficiency of SDP

The implementation of the SDP resulted in a significant reduction in the total expenditure for treating cholecystitis in 92.9% (13/14) of the studies [9,10,11,12,13,14,15,16,17,18,19,20,21], with a maximum reduction of half [13,21]. This demonstrated that SDP was effective in controlling cost (Table 1). However, the cost of treating cholecystitis without surgery was not significantly reduced after the implementation of SDP in some studies [17], suggesting that surgical instruments and consumables are the primary target for cost control, rather than examinations or medications.

The cost of laparoscopic cholecystectomy varied significantly between regions and hospitals, ranging from 5000 to 20,000 RMB, which may be related to regional development and hospital level. For example, patients treated at tertiary hospitals may incur higher expenses for laparoscopic cholecystectomy due to the presence of more comorbidities and the need for more medical treatment during the perioperative period.

In conclusion, while the SDP is simple and effective in controlling costs for certain cases, it does not account for all potential influencing factors of hospitalization expenses for cholecystitis. Further analysis of these factors is needed to fully understand the cost-saving impact of the SDP (Table 2).

### 3.4. Factors Influencing Cholecystitis Expense in SDP

Of the six studies included, all recognized LOS and age as having significant impact on hospitalization cost. Interestingly, four studies found gender to be a factor influencing the total cost, with the cost for male patients being significantly higher than that for female patients [22,23,24,25,26]. One possible explanation is that male inpatients tend to be older and have more underlying diseases, surgical history, and complications compared to female inpatients [27]. Moreover, four studies indicated that patients with operations and complications usually tend to incur higher hospitalization costs (Table 2).

**Table 2 healthcare-11-02965-t002:** Influencing factors of cholecystitis expense in SDP.

First Author	Province	LOS			Complication			Age			Gender			Operation		
		β	β′	Sig	β	β′	Sig	β	β′	Sig	β	β′	Sig	β	β′	Sig
Jiang (2019) [28]	Shandong	5.560		0.000	0.089		0.047	0.195		0.030				−4.655		0.000
He (2015) [26]	Yunnan	0.109	0.289	0.000	0.031	0.076	0.001	0.023	0.055	0.006	0.001	0.073	0.001	−0.290	−0.294	0.000
Gu (2013) [24]	Tianjin	0.011	0.373	<0.001	0.024	0.034	0.040	0.002	0.097	<0.001	0.019	0.037	0.030	−0.285	−0.544	<0.001
Feng (2010) [23]	Beijing	1.720	0.392	0.000	0.906	0.092	0.000	0.028	0.164	0.000	0.101	0.008	0.001	1.255	0.123	0.000
Chen (2006) [22]	Beijing	0.001	0.431	0.000				0.000	0.062	0.000	0.001	0.062	0.000			
Guo (2004) [25]	Anhui	1.853	0.310	0.000				0.080	0.084	0.050	3.460	0.122	0.004			

β, the regression coefficient, reflects the impact degree of expense. β′, the standard regression coefficient, reflects the impact degree of expense without unit influence. A positive value indicates a positive correlation between the factor and hospitalization expenses, while a negative number indicates a negative correlation between the factor and hospitalization expenses. The larger the absolute value, the greater the impact of the factor on hospitalization expenses. Sig, Significance. Female was assigned a value of 1 and male was assigned a value of 2.

### 3.5. The Way of DRG Grouping Cholecystitis

In light of the factors that influence the cost of cholecystitis treatment under SDP, DRG grouping provided a comprehensive framework for reimbursing cholecystitis treatment. Combined with the above conclusions and practical factors, the current DRG uses the occurrence of complications and surgical methods as the basis for grouping cholecystitis. Cases with severe complications or complex surgery are assigned higher relative weight (RW) values, as demonstrated in Table 3, which includes the number of cases treated in 2020 at Qilu Hospital. The majority of patients fall under the category of “laparoscopic cholecystectomy without common bile duct exploration”, which is equivalent to the “laparoscopic cholecystectomy” category in the SDP system. However, when patients required intensive care or novel technology, these cases were assigned a higher RW value and therefore a higher reimbursement amount. In conclusion, DRG is a more nuanced payment system that takes into account all aspects of medical practice compared to the simple SDP system.

### 3.6. The Efficiency of DRG in Cholecystitis-Related Disorders

In this study, the impact of DRG was evaluated in the surgical department of Qilu Hospital in the treatment of cholecystitis-related disorders. The results showed that among the 16 disorders included, the total expenditure was positively correlated with LOS. Following the implementation of DRG, 11 of the disorders showed a decrease in both LOS and expenditure, while nine showed an increase. The DRG had a significant impact on reducing both LOS and expenditure in groups with lower Relative Weight (RW) values; however, the effect was less remarkable in groups with higher RW values. This may be due to the fact that patients in groups with higher RW are more likely to have underlying diseases and complications, which can skew the results when the sample population is limited (Table 4).

### 3.7. Efficiency of DRG

Among the ten results from nine studies, six showed that the post-DRG group spent less than the pre-DRG group [29,30,31,32,33,34], while four demonstrated a completely opposite outcome [30,35,36,37]. One possible explanation for this discrepancy is the difficulty in controlling for cost-influencing factors when selecting samples in DRG due to their complex nature. This can lead to significant differences in variables such as diseases, age, gender, operations, complications, and others between the pre-DRG and post-DRG groups [30,36,37]. However, two studies reversed their results showing that DRG could reduce hospitalization cost significantly after analyzing influencing factors of expenditure with univariate linear regressions and multilevel mixed effects models with robust standard errors [30,36]. Therefore, multiple statistical methods might be necessary to elucidate the role of DRG in reducing expenditures from other confounding factors. Meanwhile, seven articles proposed that DRG could reduce LOS remarkably (Table 5).

## 4. Discussion

Since the 1980s, China has been exploring the indigenization of DRGs and learning from advanced DRG experience in developed countries. The “Notice on piloting the single disease payment” issued by the Chinese Ministry of Health in November 2004 is regarded as the initial sign of China’s exploration to indigenize DRG [38]. In contract with mature DRG, SDP, which is essentially a prospective payment system, is regarded as a simplified version of DRG. The essence of SDP is that the charge of a medical institution is only related to the diagnosis of a specific disease, but not related to the actual cost to treat the case, the so-called “ceiling price for a single disease” [39].

Our study, taking cholecystitis as an example, found that the implementation of SDP significantly reduced the expenditure and length of hospital stay for cholecystitis. Moreover, three publications reported that patients were satisfied with the SDP system [12,19,20]. and one study reported that SDP raised medical efficiency by increasing inpatient turnover [14]. The results of the analysis of the factors influencing cholecystitis expense in SDP also showed the relevance of expenditure with LOS, gender, age, complications, and operations, which provided a powerful basis for the criteria for subsequent grouping of DRG.

However, as defined, SDP is just a simplified and transitional version of DRG, not a genuine DRG system. The limitations of SDP are evident: its coverage of disease spectrums is limited, it ignores other important factors such as the severity of disease, age, gender, and complications, and it is only concerned with the first diagnosis, with no uniform standards [40,41]. Hence, further innovation and development are needed for a localized DRG system.

In 2008, the Beijing Medical Insurance Association developed the Beijing version of DRG (BJ-DRG), which was the first genuine localized DRG system. Since then, various DRG systems have been piloted in different provinces. Currently, there are four widely used and influential DRG systems in China, each with its own features. The BJ-DRG primarily focuses on cost control, CN-DRG emphasizes medical performance evaluation and quality supervision, C-DRG groups diseases innovatively based on the full spectrum, and CHS-DRG is a culmination of all the versions (Table 6).

In this study, average hospitalization cost is remarkably reduced after the implementation of DRG in most research. At the same time, there is no significant difference between the post-DRG group and the pre-DRG group in healthcare quality and safety [30,31,32,33,34,35,36,37]. This finding suggests that DRG can reduce medical costs without affecting medical quality and safety. Two studies also show that patient satisfaction was significantly improved after implementing DRG [30,35]. However, in one study, the low-risk-mortality was higher in the post-DRG group (0.24%) than in the pre-DRG group (0.08%) [30]. In addition, several researchers proposed that some medical institutions have insufficient compensation due to the large number of critically ill patients, which may be the consequence of higher mortality in DRG group [30,33,37]. Moreover, two studies utilizing Data Envelopment Analysis (DEA) revealed that DRG was not effective enough in improving medical efficiency, especially in clinical departments [42,43]. At the same time, DRG relies on high-quality medical records but paradoxically lacks universal standards, making it arduous to popularize in hospitals lacking medical resources.

In order to overcome the regional differences and difficulties in the application of DRG, a new payment convenient to apply in urban hospitals is urgent. Combining the characteristics of big data and traditional DRG, in 2020, the Chinese National Healthcare Security Administration creatively proposed the concept of DIP [44]. Compared with DRG, DIP takes advantage of Big Data to divide different diseases with different surgeries into different groups, which means there is only one disease with or without one specific operation in one group (Table 7). This alteration makes high-quality medical records no longer necessary in payment and enables DIP implementation in remote hospitals lacking healthcare resources.

However, there were several limitations in this study. Firstly, some unpublished studies may have been missed despite identifying and including the largest number of studies. Secondly, although the utilization of DEA provided a more comprehensive perspective for studying the impact of DRG on medical efficiency, the lack of available data affected the accuracy of the results. Thirdly, the details of the components of DRG payment policy and related context were underreported in the included studies, which might limit the interpretation and application of research findings. Lastly, as DIP is a newly proposed concept, there is currently a scarcity of research that evaluates its efficiency. Future studies should focus on collecting relevant data in these areas.

## 5. Conclusions

At present, the implementation of SDP, DRG, and DIP has spread across both urban and rural areas in all provinces and municipalities in China. These systems are different stages of the indigenization of DRG and complement each other. SDP provides a convenient payment system to control certain disease expenses. DRG, proven effective in many developed countries, displays its potent ability to manage medical costs but is limited in less-developed regions. DIP provides a new method of reducing medical insurance costs by adapting to DRG localization based on the national conditions of developing countries in China, although its efficiency has not yet been evaluated. At different times, each of these three DRG systems fulfills a distinct role and contributes significantly to the achievement of medical cost containment. For developing countries, replicating the DRG system used in developed countries can be challenging due to limited primary care resources. However, the experience of multiple DRG variants complementing each other in China provides a valuable experience in controlling medical expenditure.

## Figures and Tables

**Figure 1 healthcare-11-02965-f001:**
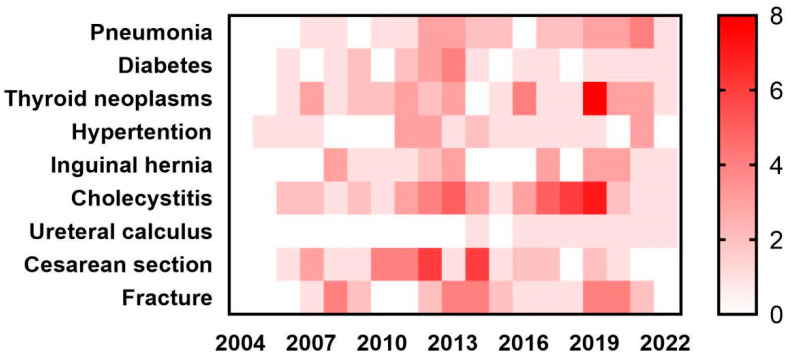
Distribution of SDP studies by years and diseases.

**Figure 2 healthcare-11-02965-f002:**
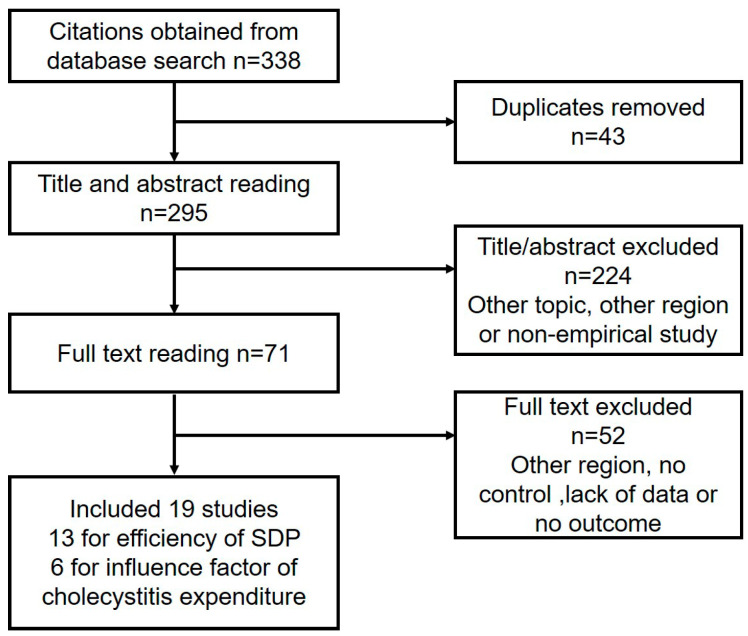
Flow chart of SDP study selection.

**Figure 3 healthcare-11-02965-f003:**
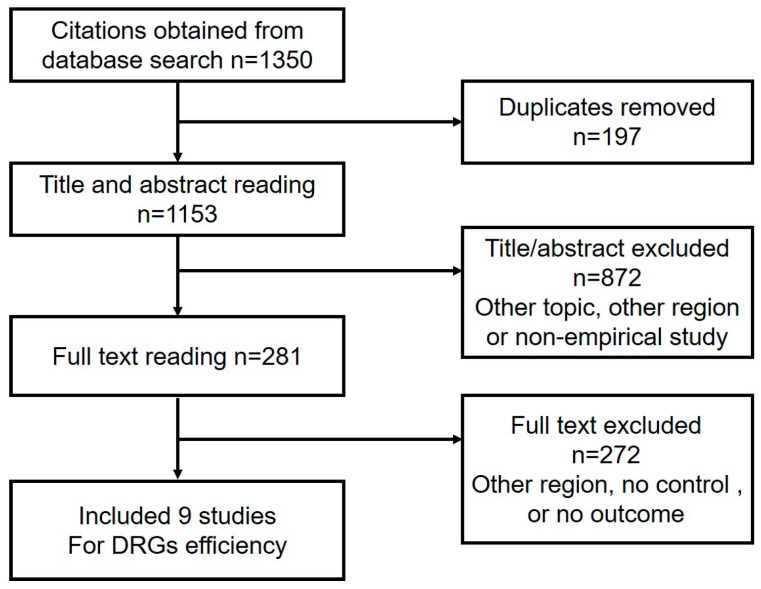
Flow chart of DRG study selection.

**Table 1 healthcare-11-02965-t001:** Efficiency of SDP.

First Author	Disease/Operation	Province	Number		Total Expenditure (RMB)			LOS (Day)		
			Post- SDP	Pre- SDP	Post- SDP	Pre- SDP	*p* Value	Post- SDP	Pre- SDP	*p* Value
Tang (2019) [9]	Laparoscopic cholecystectomy	Shandong	266	202	13,858.32	21,699.15	0.00	4.70	8.31	0.02
Sun (2019) [12]	Laparoscopic cholecystectomy	Hunan	75	260	10,533.79	13,355.16	<0.01	5.00	6.00	<0.01
Zhou (2018) [17]	Cholelithiasis with acute cholecystitis (Operation)	Yunnan	27	30	5859.34	6264.37	0.05	6.49	7.47	0.16
Zhou (2018) [17]	Cholelithiasis with acute cholecystitis (No Operation)	Yunnan	44	51	3473.20	3537.37	0.66	5.48	6.06	0.03
Zhang (2018) [18]	Laparoscopic cholecystectomy	Liaoning	3864	4037	21,738.00	24,304.00	<0.01	10.10	10.23	0.35
Chen (2016) [10]	Cholelithiasis with acute cholecystitis	Anhui	755	940	5550.80	6026.00	0.00	6.55	9.04	0.00
Sun (2015) [11]	Laparoscopic cholecystectomy	Jiangsu	23	23	9372.58	11,653.52	<0.05	7.13	9.85	<0.05
Wang (2013) [13]	Laparoscopic cholecystectomy	Shandong	50	50	4073.35	9887.13	<0.01	5.00	13.00	<0.01
Yu (2012) [15]	Laparoscopic cholecystectomy	Jilin	88	119	12,534.38	16,716.18	<0.01	5.00	7.00	<0.01
Kong (2011) [14]	Laparoscopic cholecystectomy	Jiangsu	100	100	3989.34	6321.88	<0.05	4.40	6.80	<0.05
Su (2011) [21]	Laparoscopic cholecystectomy	Hunan	383	516	5017.84	11,355.93	<0.01	8.20	12.85	0.00
Liu (2009) [19]	Laparoscopic cholecystectomy	Jilin	139	163	8100.00	9900.00	<0.01	5.80	8.20	<0.01
Jiang (2009) [18]	Laparoscopic cholecystectomy	Hubei	95	91	5483.83	6470.44	<0.05	8.20	10.50	<0.05
Ren (2006) [20]	Laparoscopic cholecystectomy	Hebei	118	118	4390.00	5533.00	<0.05	4.00	7.00	<0.05

**Table 3 healthcare-11-02965-t003:** DRG grouping scheme of cholecystectomy implemented in Qilu hospital.

DRG Code	DRG Group	RW	Number of Cases
HS10B	Laparoscopic cholecystectomy without common bile duct exploration	0.95	2006
HS10C	Laparoscopic cholecystectomy, day surgery	0.95	2
HS09C	Open cholecystectomy without exploration of common bile duct or severe complications	1.66	82
HS10A	Laparoscopic cholecystectomy with exploration of common bile duct	1.75	78
HS10 + HO03	Complex treatment of laparoscopic cholecystectomy + ERCP	1.77	52
HS09A	Open cholecystectomy with exploration of common bile duct	2.05	16
HS09B	Open cholecystectomy with exploration of common bile duct and severe complications	2.05	8

**Table 4 healthcare-11-02965-t004:** The efficiency of DRG in cholecystitis-related disorders.

DRG Groups	RW	Number		LOS (Days)			Total Expenditure (RMB)		
		Pre- DRG	Post- DRG	Pre- DRG	Post- DRG	Difference	Pre- DRG	Post- DRG	Difference
Major surgery of biliary tract, with malignant tumor	2.54	28	50	18.54	16.52	2.02	80,940.56	80,544.06	396.50
Complex operations of ERCP, with severe complications and concomitants	2.16	18	49	9.00	9.80	−0.80	29,208.42	33,511.80	−4303.38
Other minor hepatobiliary and pancreatic surgery	2.09	37	124	9.41	5.07	4.34	18,188.37	13,832.44	4355.92
Percutaneous bile duct therapeutic procedure	1.80	69	68	10.19	8.73	1.46	26,044.01	20,340.31	5703.70
Laparoscopic cholecystectomy, with exploration of common bile duct	1.75	16	20	14.00	11.70	2.30	43,822.88	44,061.44	−238.56
Laparotomic cholecystectomy, without exploration of common bile duct, without severe complications and concomitants	1.66	18	17	13.56	15.18	−1.62	48,591.35	54,390.31	−5798.96
Major surgery of biliary tract, without malignant tumor, with moderate complications and concomitants	1.46	10	13	14.20	13.77	0.43	43,288.88	53,139.44	−9850.56
Complex operations of ERCP, without severe complications and concomitants	1.38	81	149	8.37	7.23	1.14	25,374.83	24,181.43	1193.40
Major surgery of biliary tract, without malignant tumor, without complications and concomitants	1.15	44	43	12.82	10.00	2.82	38,133.45	36,957.94	1175.50
Diagnostic operations of hepatobiliary system without severe complications and concomitants	1.06	21	45	7.29	6.91	0.38	12,986.78	13,025.49	−38.71
Laparoscopic cholecystectomy, without exploration of common bile duct	0.95	540	804	7.49	6.21	1.28	23,612.10	22,852.57	759.53
Malignant tumor of liver, gallbladder, or pancreas, without severe complications and concomitants	0.85	154	47	9.92	7.63	2.28	27,536.63	15,523.52	12,013.11
Acute biliary tract disease, with complications and concomitants	0.81	14	16	7.89	7.69	0.20	16,830.59	16,415.66	414.93
Chronic biliary tract disease, with complications and concomitants	0.70	18	22	6.66	5.32	1.34	10,504.02	8408.61	2095.41
Acute biliary tract disease, without complications and concomitants	0.64	30	25	7.57	5.84	1.73	15,092.63	8649.21	6443.42
Chronic biliary tract disease, without complications and concomitants	0.40	74	74	7.09	5.12	1.98	8267.54	6602.42	1665.12

Difference is obtained by subtracting the value of “post-DRG” group from the value of “pre-DRG” group.

**Table 5 healthcare-11-02965-t005:** The efficiency of DRG.

First Author	DRGs	Province	Number		TE (RMB)			LOS (Day)		
			Post- DRG	Pre- DRG	Post- DRG	Pre- DRG	*p* Value	Post- DRG	Pre- DRG	*p* Value
Liu (2020) [29]	Local DRG	Henan	9381	10,176	22,738	25,837	<0.05	13.30	15.80	<0.05
Meng (2020) [30]	CN-DRG	Guangdong	439,114	300,538	5301	5633	<0.001	8.85	8.81	<0.001
Meng (2020) [30]	C-DRG	Guangdong	232,255	300,538	5681	5633	<0.001	8.70	8.81	<0.001
Zhou (2022) [33]	Local DRG	Yunnan	23,887	165,261	14,819	17,435	<0.001	N/A	N/A	N/A
Tang (2020) [35]	CN-DRG	Beijing	5982	2871	20,674	19,960	0.139	9.98	10.14	0.044
Zhou (2021) [37]	CHS-DRG	Zhejiang	24,728	18,355	10,125	9720	<0.01	7.00	7.00	<0.01
Zhou (2019) [36]	Local DRG	Liaoning	995	2165	19,048	17,510	<0.01	N/A	N/A	N/A
Zhang (2022) [32]	Local DRG	Hubei	62	65	12,437	18,032	0.058	8.19	8.80	0.058
Wu (2021) [31]	Local DRG	Sichuan	12,019	11,533	8658	10,951	<0.05	8.78	10.66	<0.05
Zou (2020) [34]	C-DRG	Guangdong	237	234	11,739	12,278	0.725	4.87	5.34	0.019

**Table 6 healthcare-11-02965-t006:** Popular DRG versions in China.

	R&D Department	Year Published	Number of Groups (ADRG/DRGs)	RW	Character
BJ-DRG	Beijing Medical Insurance Association	2008	393/752	Resource consumption	First localized DRG in China Focuses on expense control
CN-DRG	Beijing Municipal Health Commission	2014	415/804	Total expense of disease consumption	Focuses on medical performance evaluation and quality supervision
C-DRG	National Health Commission of the People’s Republic of China	2015	455/958	Total cost of disease consumption	Innovatively groups based on all disease spectra and CCHI
CHS-DRG	National Healthcare Security Administration	2019	367/618	Total cost of disease consumption	Refers to and integrates all above versions

Adjacent-DRG (ADRG), the secondary classification in DRG system, groups cases mainly according to diagnosis and operations based on the broad categories verified by primary classification of major diagnostic categories (MDCs), while the tertiary classification diagnosis-related groups (DRGs) subdivides cases on this basis.

**Table 7 healthcare-11-02965-t007:** Comparison of SDP, DRG and DIP.

	SDP	DRG	DIP
Payment	Prospective payment	Prospective payment	Prospective payment
Data source	Medical records	Medical records	Medical records
Grouping objects	Simple diseases with no comorbidities and complications	All diseases	All hospitalized patients
Payment standard	Categories of disease	Groups and Relative Weights	Groups and budget point value
Standardized	No	Only MDC and ADRG are standardized	Yes
Grouping pathway	N/A	Induction	Exhaustive method
Grouping basis	N/A	Clinical pathway	Clinical data
Grouping refinement	N/A	Similar diagnoses and operations in one group	Different diseases and operations in different groups
Intra-group differences	N/A	Significant	Insignificant
Number of groups	N/A	Usually less than 1000	More than 10,000
Requirement of medical record quality	Low	High	Low
Applying difficulty	Easy	Hard	Easy
Usage of big data	No	No	Yes

MDC: Major Diagnostic Category, ADRG: Adjacent-DRG.

## Data Availability

Data are not available due to hospital privacy policies.

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
