# Peer review of "Statistical Insight into China’s Indigenous Diagnosis-Related-Group System Evolution"

_healthcare, 2023, doi:10.3390/healthcare11222965_

Round 1
Reviewer 1 Report
Comments and Suggestions for Authors
This is an interesting study evaluating China’s SDP and DRG systems through both literature review and empirical analyses. I have two comments that may strengthen this paper:
1. It’d be helpful to the readers if the authors briefly explain and compare SDP, DRG, and DIP in the introduction. Section 3.5 touches on SDP vs. DRG a bit, but providing an overview in the intro instead would give readers a better idea of the big picture.
2. In Section 3.6, the authors characterize the analysis as comparing experimental group and control group. However, this is really a before-and-after comparison. I’d use the terms “pre-DRG” and “post-DRG” instead of “EG” and “CG” in Table 4 and the text. Did the literature reviewed in other sections also use simple before-and-after comparisons? I’d also change the terms in those sections if that’s the case.
Comments on the Quality of English LanguageThis is mostly well written. Some minor edits can be made to improve the writing.
Reviewer 2 Report
Comments and Suggestions for Authors
The study with a single disease-based prospective payment experiment is very unique in China. The paper is well written and offers interesting results. Several issues pertaining to the methodology should be addressed.
First, it is unclear how costs are compiled. Are they charge-based reported by hospitals?
Second, it is important to include the severity of the disease in the single disease approach.
Third, Data Envelopment Approach (DEA) could be considered when the productivity or technical efficiency measurement is being considered.
Fourth, although the sample size is relatively large, it is necessary to validate the consistency of DRGs or classifications. Furthermore, the limitations of the proposed approach should be carefully addressed.
In conclusion, I recommend that the paper be amended and moved forward for further consideration by the editorial panel.
Comments on the Quality of English LanguageOK
Reviewer 3 Report
Comments and Suggestions for Authors
1. The introduction doesn't provide sufficient background and doesn't include all relevant references. References should bee actualized. Too much DRG historical background too little reference to how it works in other countries.
2. Research design are not clear for me. Please, explain more precisious. Especially concerning secondary data and database searches. I can't see connection between primary and secondary data.
3. Table two is not celar - can you explain behind?
4. Conclusion aren't supported by the results.
Round 2
Reviewer 1 Report
Comments and Suggestions for Authors
The authors have adequately addressed my comments.
Author Response
Thank you very much for all your valuable suggestions.
Reviewer 2 Report
Comments and Suggestions for Authors
The revised paper has clarified a majority of the critical points noted in the previous review. Two minor issues could be further clarified by authors as follows:
1. Concept of efficiency: This term should be clearly defined in the introduction section since there are three different aspects of efficiency that could be explored, such as the cost efficiency, process efficiency, and technical efficiency.
2. Efficacy: This term was also used in the paper, but authors should be aware of the implication of efficacy. It implies that the outcomes generated from a RCT-based clinical study. The literature has shown that there is difference between efficacy and effectiveness.
If authors are able to clarify the above issues, I recommend that the revised paper be moved forward for further consideration by the editorial board.
Author Response
Thank you for your insightful feedback on our manuscript. We genuinely appreciate your diligent review and constructive comments. We also feel very sorry for all the inconvenience brought to you because we confused the concept of “efficiency” and “efficacy”. In fact, “efficacy” is not proper to be used in this article and it was used by mistake when we polished language. In response to your concerns and to enhance clarity, we replaced all “efficacy” by “efficiency”. The change can be found in line 298 and line 284.
As for the concept of “efficiency”, it refers to two different concepts in this article. One refers to the efficiency of all aspects including cost effectiveness and medical efficiency. One only refers to the concept of medical efficiency. To extinguish between these two concepts, we defined the concept of “efficiency” including both “cost effectiveness” and “medical efficiency” in the first sentence in introduction from line 31 to line 34. “Originating from Yale University, Diagnosis Related Groups (DRG) was first implemented in the United States in 1983 and is considered a highly effective payment system to improve efficiency, including controlling hospitalization costs and increasing medical efficiency.”
Besides, we also replaced all concept referring to medical efficiency by “medical efficiency” in line 64, 68, 75 and 234.
All changes were highlighted with red color in the manuscript.